# Carbon Dioxide Adsorption by a High-Surface-Area Activated Charcoal

**Ahmed S. Ahmed** [1,*], **Mohammed Alsultan** [1,*], **Assim A. Sabah** [1] **and Gerhard F. Swiegers** [2,*]

1    Department of Science, College of Basic Education, University of Mosul, Mosul 41002, Iraq
2    Intelligent Polymer Research Institute and ARC Centre of Excellence for Electromaterials Science, University of Wollongong, Wollongong, NSW 2522, Australia
*    Correspondence: ahmed.saadallah@uomosul.edu.iq (A.S.A.); mohamadfkaleel@uomosul.edu.iq (M.A.); swiegers@uow.edu.au (G.F.S.)

**Abstract:** An activated carbon (AC) with a high surface area of 4320.7 $m^2/g$ was synthesized via the chemical and thermal processing of walnut residues. The resulting activated charcoal was characterized by various techniques, including Brunauer–Emmett–Teller (BET) surface area analysis, scanning electron microscopy (SEM), and IR spectroscopy. The prepared activated carbon was studied for its capacity to adsorb $CO_2$ gas. When exposed to $CO_2$ for 60 min in a closed chamber at atmospheric pressure at 25 °C, the adsorption of a notable 301.1 mmol $CO_2$ (13.25 g $CO_2$) per gram of activated carbon was observed.

**Keywords:** charcoal; activated charcoal; walnuts; carbon dioxide; adsorption

## 1. Introduction

Carbon dioxide ($CO_2$) is considered a major pollutant gas. $CO_2$ emitted to the atmosphere due to the burning of fossil fuels is believed to facilitate an environmentally undesirable greenhouse effect, leading to global warming. Significant efforts are being undertaken by nations, companies, and organizations to reduce their $CO_2$ emissions. Alternative energy sources that reduce $CO_2$ emissions are significantly increasing. This study also examines the capture of $CO_2$ from the air. The design of materials that can adsorb $CO_2$ has been widely investigated in the last few decades [1].

Activated carbon (AC) is a black material that is solid, tasteless and has a high porosity. It is part of a large family of coal materials that do not have a specific chemical composition. Activated carbon can be prepared from several different materials (natural carbon raw materials such as walnut shells, wood, coal, and pistachio shells) using different chemical approaches. The composition of activated carbon involves functional groups that provide an active location in the adsorption phenomenon. Among these are carboxylates are carboxylic hydroxide, phenolic, carbonyl, and lactone groups.

In general, when treated with acidic chemical materials, activated carbon has acidic properties. These could be promising for the adsorption of basic gases, such as ammonia. Due to the presence of active oxygen on the surface of the activated carbon (for example, carboxyl, ketone, and phenol groups). In contrast, when activated carbon is treated with basic materials, it may adsorb acidic gases such as sulfur dioxide due to presence of carbonyl chromene groups, as shown in Figure 1.

Activated carbon is a porous carbonaceous material whose chemical and physical properties have led to its widespread use in environmental applications, including the treatment of wastewater (to remove color), removal of pollutant gases, desulfurization of petroleum, and adsorption of antibiotic drugs [2–5].

**Figure 1.** Some of the active groups that may be present on an activated carbon surface.

Activated carbon (AC) has also been studied as an adsorbent of carbon dioxide ($CO_2$), with activated carbon derived from walnut residues (AC-W) proving particularly useful in this respect. The method of producing the AC-W may significantly affect its capacity to adsorb $CO_2$. For example, Chomiak and colleagues demonstrated $CO_2$ adsorption of up to 7.2 mmol $CO_2$ per g AC-W at 1 bar and 18.2 mmol/g at 30 bar using AC-W prepared by soaking with KOH in the ratio 1:2 (C:KOH) at 800 °C [6]. Lewick and Seraft prepared AC-W by soaking for 1 h in the ratio 1:1 (C:KOH), followed by drying for 19 h. The resulting AC-W adsorbed 4.36 mmol of $CO_2$/g. However, when soaked in the ratio 1:0.75 (C:KOH) for 24 h and then dried at 500 °C for 5 h, the AC-W was reportedly able to adsorb a remarkable 920.6 mmol/g AC-W at atmosphere pressure, which appears to stand as the highest recorded value to date [7]. The origin of these differences derives from the surface area and the activated functional groups in the AC, which may significantly affect its adsorptive capacity [2].

Activated carbon consists of an organized composition of fine pores of variable size (150 Å to 5 Å), consisting of irregular layers that are well separated (6.3 Å) [8–10]. The structure of activated carbon involves an arrangement of microscopic, mesoscopic, and macroscopic pores. Table 1 broadly lists the properties and the pore types. The activity of an activated carbon for $CO_2$ adsorption depends on the nature of the material, including its pore structure, and the method of its preparation, which affects the surface of the activated carbon, especially its porosity. Surface areas of up to 3000 m²/g are most common [11,12].

**Table 1.** Classification and characteristics of activated carbon pores [5,13].

|  | Volume—Mass (g/mL) | Specific Surface (m²/g) | Diameter (nm) | Pore Types |
|---|---|---|---|---|
| Micropores | 0.2–0.6 | 600–1500 | <2 | Fine pores |
| Mesopores | 0.02–0.1 | 20–70 | 2–50 | Medium pores |
| Macropores | 0.2–0.8 | 0.5–2 | >50 | Large pores |

Activated carbon is available in powder form or in granules. In fine powder form, it typically has a particle size of less than 100 μm, with an average diameter of 15–25 μm, a large outer surface and very fast adsorption velocity. Fine-powder activated carbons can be used to change the color of oils, fats, wine, sugars, and many other organic liquids. In granular form, activated carbon may have a particle size of >1 mm, a small diameter, a large inner surface area, and a relatively small outer surface. As a result, the 'spread'

phenomenon within the pore is highly important in the adsorption process. Granules are usually used for wastewater treatment, gases, and harsh steam treatments [14–16].

The speed of the adsorption processes depends on physical and chemical factors. Physically, the particle size of the activated carbon plays a crucial role in the adsorption speed. Smaller particles increase adsorption speed due to more rapid 'spread' reaching the center [17,18]. Other physical factors affecting adsorption speed involve the resistance of the activated carbon to corrosion, pressure, depletion, and vibration. This mostly depends on the raw materials and their level of activation [19–21]. The most common chemical factor is the production of ash during the preparation of the activated carbon. Ash is the inorganic, amorphous and unusable part of activated carbon. It often contains calcium salts and mineral oxides; therefore, the lower the ash rate, the more strongly adsorbent the activated carbon [22].

In this study, we describe the preparation of an activated carbon with a high specific surface area of 4320.7 $m^2/g$. The activated carbon was examined for $CO_2$ gas absorption in a closed chamber system and found to adsorb around 300 mmol per gram of activated carbon over 60 min at atmospheric pressure.

## 2. Materials and Methods

### 2.1. Raw Materials

Walnut shells (25 g) were collected from the local area in Mosul, Iraq and used as the raw carbon material.

### 2.2. Equipment

X-ray photoelectron spectroscopy (XPS) was carried out using a model PHI660 (Rodgau, Germany) with monochromatic Mg K$\alpha$ X-ray as a single-beam source. FTIR spectroscopy employed a Shimadzu model IRTracer-100. Sigma-Aldrich tables of IR absorptions were used for analysis. Scanning electron microscopy (SEM) was carried out using TESCAN (Brno, Czech Republic). The oven was a Carbolite model operated at 1100 °C.

### 2.3. Preparation of Activated Carbon

Activated carbon was prepared from Iraqi walnut shells. The preparative procedure involved two steps, namely pyrolysis followed by physical/chemical activation. In the pyrolysis step, the carbonaceous raw materials were typically heated to high temperatures in the range 400–600 °C in an isolated environment. This removed the heterogeneous elements (oxygen, hydrogen, nitrogen), which departed as gases, resulting in a solid carbon material with a high porosity [23]. In the physical activation step, an oxidizing agent such as air, water vapor, or carbon dioxide at high temperature (850–1100 °C) was introduced. Thereafter, the AC had a lower mass; the proportion indicated the activation rate [24]. Chemical activation occurs when the charred substance is treated, typically with phosphoric acid ($H_3PO_4$), zinc chlorine ($ZnCl_2$), potassium hydroxide (KOH) or sulfuric acid ($H_2SO_4$). The substance is then exposed to a low temperature (compared with physical activation) in order to reorganize the structure of the activated carbon. After the reactions are complete, the AC material may usually be washed with distilled water to remove all traces of the remaining chemicals [24,25].

A quantity of walnut shells was collected and crushed into small fragments of about 1 cm size and washed with distilled water to remove all impurities. After first drying in air, and then in an oven at 60 °C for 24 h, the raw material was ground and filtered through a sieve with dimensions of 125–250 μm. The collected sample was washed with distilled water several times and then dried in a thermal drier at 70 °C for 5 h.

The resulting charcoal raw material (25 g) was placed in a crucible and soaked with concentrated KOH (dissolved in a minimum volume of water) in the ratio (*w/w*) (1 C:0.75 KOH). The crucible was placed in an oven (Carbolite-CWF 1100, Neuhausen, Germany) at 500 °C for 1 h, whereafter it was allowed to cool to room temperature. The overall procedure is depicted in Figure 2.

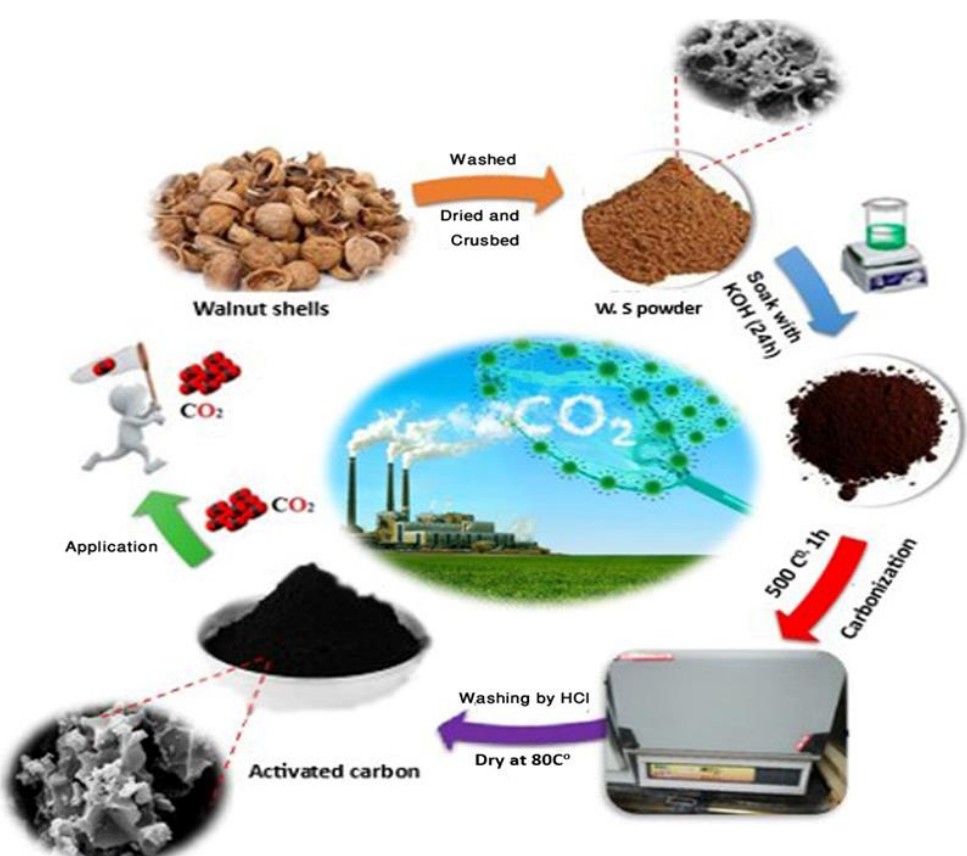

**Figure 2.** Process steps to prepare the activated carbon.

### 2.4. Measuring the Surface Area of Activated Carbon through Iodine Number

The iodine number was used to measure the surface area of the activated carbon, and it was 4126 $m^2g^{-1}$. When compared with the BET measurement, whose value was 4320 $m^2g^{-1}$, it was noted that the values were close to each other, and this supports the laboratory result obtained by manually measuring iodine number via calibration with thiosulfate.

To carry out the experiment in more detail, 1 gm of dry activated carbon was weighed and placed in a volumetric flask with a capacity of 100 mL, and then 10 mL of 5% HCl solution was added and boiled for 15 min (to remove any material that might interfere with the thiosulfate later during calibration). The solution was left to cool to laboratory temperature, and then 100 mL of 0.1 N iodine solution (prepared from dissolving 12.7 gm iodine with 19.1 gm potassium iodide in a liter of water) was added. Then, the resulting mixture was shaken via ultrasound for 30 s, filtered, and the first 20 mL of the filtrate was discarded. Then, 50 mL of the filtrate was taken and calibrated with thiosulfate (0.1 N sodium thiosulfate solution was prepared by dissolving 24.8 gm sodium thiosulfate with 0.1 gm sodium carbonate in a liter of distilled water) until it turned a pale yellow color. Then, a few drops of starch indicator was added, carburation continued until the color disappeared, and the number from the burette was recorded.

### 2.5. Measuring the Surface Area of Activated Carbon via Methylene Blue

The methylene number measured the surface area and was 83 mg/g, representing the adsorption rate of methylene dye per gram of coal. This result was confirmed after the disappearance of the methylene blue color and is a measure of the absorption capacity of activated carbon.

Next, 0.1 gm of dried activated carbon was taken and placed in a volumetric flask, and 50 mL of methyl blue dye was added at a concentration of 20 ppm (prepared by

dissolving 0.02 gm of the dye in a liter of distilled water). The solution was shaken for two hours in an electric shaker until the color of the methyl blue dye disappeared. Another quantity of 50 mL of dye solution was added, and the addition process continued until the color stabilized.

The surface area from methylene blue was calculated using the following equation:

$$Z = y \times V/1000 \tag{1}$$

where Z is the surface area of AC, y = AC adsorbed by methylene blue (and equal 20-X), and X is the concentration of the total volume produced at color stability, which is represented by the graphic curve of concentration versus absorbance in the absorbance plot shown in Figure 3.

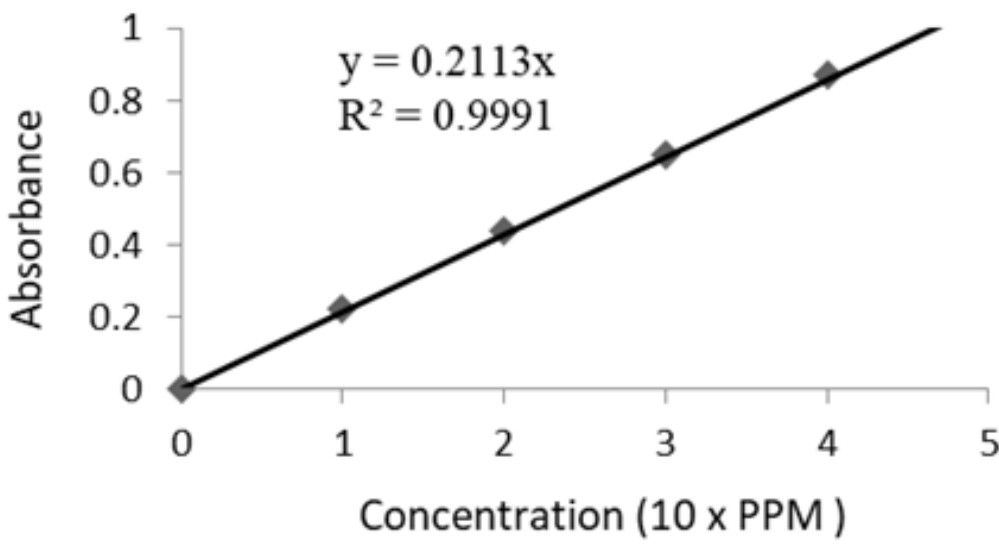

**Figure 3.** Standard curve of absorbance to calculate the surface area of AC via the methylene blue dye method.

*2.6. Humidity Measurement of Activated Carbon*

A 1 gm sample of dry activated carbon was openly placed in a laboratory for 24 h and thus exposed to air and atmospheric humidity, after which the charcoal was weighed. The weight difference represents the amount of moisture gained by the coal, and the moisture ratio is then calculated via the following equation (ASTM D2247-15)):

$$\% = (part/whole)100 \tag{2}$$

*2.7. Density Measurement of Activated Carbon*

A volumetric vial of 5 mL capacity was weighed empty and then filled with activated carbon to mark the volumetric vial. Alternatively, it was filled with activated carbon, and then the volume of the vial was subtracted to calculate the density via the following equation (ASTM D854-02):

$$d = mass/volume \tag{3}$$

*2.8. Ash Measurement*

A ceramic lid containing 1 gm of activated carbon was placed in an electric oven and heated to a temperature of 1000 °C for an hour to burn the charcoal, after which the remaining carbon in the lid was weighed. Additionally, the difference between the two weights represents the percentage of ash, and the ratio was calculated using the following equation (ASTM D2866-94).

### 2.9. Thermal Gravimetric Analysis TGA of AC

The thermal stability of activated carbon was determined using METTLER TOLEDO device and its thermal analysis program: STARe Evaluation Software version 15.01 (2018).

### 2.10. Loss of Biomass

The loss rate of biomass during the preparative procedure can be calculated using the following equation:

$$mass\ loss = \frac{W_i - W_f}{W_i} \times 100 \tag{4}$$

where $w_i$ represents the primary weight (prior to the pyrolysis step) and $w_f$ is the final weight of the sample (after activation).

### 2.11. $CO_2$ Adsorption Studies

As depicted in Figure 4, a simple device was developed to release $CO_2$. Concentrated hydrochloric acid (10 mL) was added to calcium carbonate (4 g) to release $CO_2$ gas. The gas was passed through a drying bed at 25 °C to remove moisture. The weight of generated $CO_2$ was measured in an air-evacuated graduated cylinder placed on a sensitive balance. In a typical set of experiments, around 1.596 g of $CO_2$ gas was produced on each occasion. This was then allowed to flow into a separate closed vessel containing 0.100 g of the activated carbon made from walnut shells (vessel dimensions: 8 cm × 8 cm × 8 cm). Contact between the activated carbon and $CO_2$ gas at atmospheric pressure was allowed for various time intervals, after which the non-adsorbed $CO_2$ was released. The weight of the activated carbon before and after $CO_2$ adsorption was determined and compared.

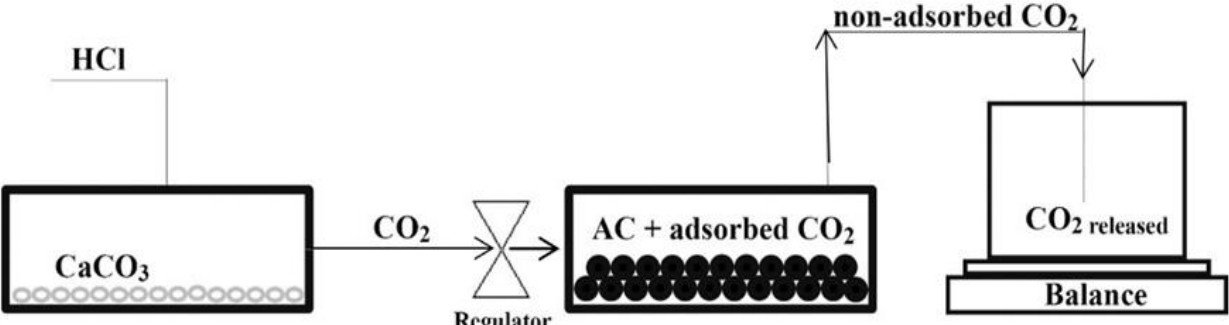

**Figure 4.** Schematic depicting the device that was developed to study adsorption and release of $CO_2$ by the activated carbon.

## 3. Results and Discussion

### 3.1. Characterization of the Activated Carbon

Activated carbon prepared (as described above) lost around 76% of its primary weight due to the formation of volatile gases during the thermal pyrolysis step. The output of the activated carbon from the raw material was therefore around 24%.

Figure 5 shows the infrared spectrum of the activated carbon before (Figure 5a) and after (Figure 5b) pyrolysis and activation. A clear decrease in the hydroxyl group (OH⁻) at 3352 cm$^{-1}$ can be seen; this is due to the pyrolysis process, which results in water being lost from the raw material. The peaks at 2920 cm$^{-1}$ and 2844 cm$^{-1}$ in the raw walnut shells belong to the aliphatic bonds (C-H) in the $CH_2$ and $CH_3$ groups in the cellulose structure. As shown in Figure 5b, these peaks largely disappeared, indicating that the cellulose structure was significantly modified by the pyrolysis process. In addition, two other peaks that were removed were located between 1350 cm$^{-1}$ and 1450 cm$^{-1}$ and belong to the $CH_2$ and $CH_3$ aliphatic groups in the structure of crystalline cellulose. A large amount of hydrogen was therefore lost during preparation.

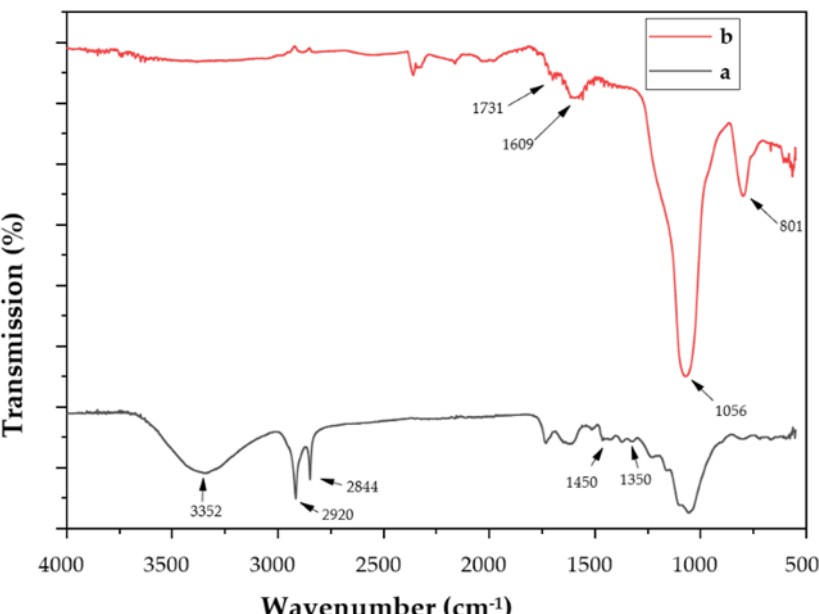

**Figure 5.** Infrared spectrum of (**a**) walnut shells (raw carbon material) before pyrolysis and activation and (**b**) activated carbon after pyrolysis and activation.

The peak at 1731 cm$^{-1}$ corresponds to the expansion vibration of the carbonyl groups (C=O), which decreased slightly, indicating that many aliphatic and aromatic bonds were broken as a result of the chemical activation and the removal of the volatile substances. The peak at 1609 cm$^{-1}$ indicates the expansion of the (C=C) aromatic group, which slightly increased in activated charcoal. Finally, the peak at about 1056 cm$^{-1}$ was caused by the presence of C-OH or C-C. A peak at 1800 cm$^{-1}$ in activated charcoal refers to C-H expansion in aromatic compounds.

### 3.2. AC Surface Structure

Scanning electron microscopy (SEM) was carried out to compare the samples before and after carbonization and activation in terms of surface area, as well as the number of pores and cracks. SEM images taken of the pure activated carbon show a large number of pores and cracks on the surface, indicating the presence of internal caves. High levels of pores and cracks enhance the surface area of the activated carbon, which may be expected to increase the adsorption of $CO_2$ gas. Elemental analysis was performed using EDX data. The ratio of oxygen increased from 3.6 to 31.5% while carbon decreased from 91.5 to 65.45%. The results of BET measurements corresponded with the SEM images and confirmed the higher surface area.

### 3.3. Surface Area Measurement (BET)

The specific surface area was measured using Brunauer–Emmett–Teller (BET) nitrogen gas adsorption at 77 K with following formula:

$$\frac{1}{Q\left(\frac{P_o}{P}-1\right)} = \frac{C-1}{Q_m C}\left(\frac{P}{P_o}\right) + \frac{1}{Q_m C} \tag{5}$$

where $P$ and $P_o$ represent the gas pressure and saturation pressure, respectively; $Q_m$ is the quantity required to satisfy a single layer in cm$^3$/g; and $C$ is a constant. Figure 6 displays the BET adsorption isotherm obtained and Table 2 summarizes the obtained values.

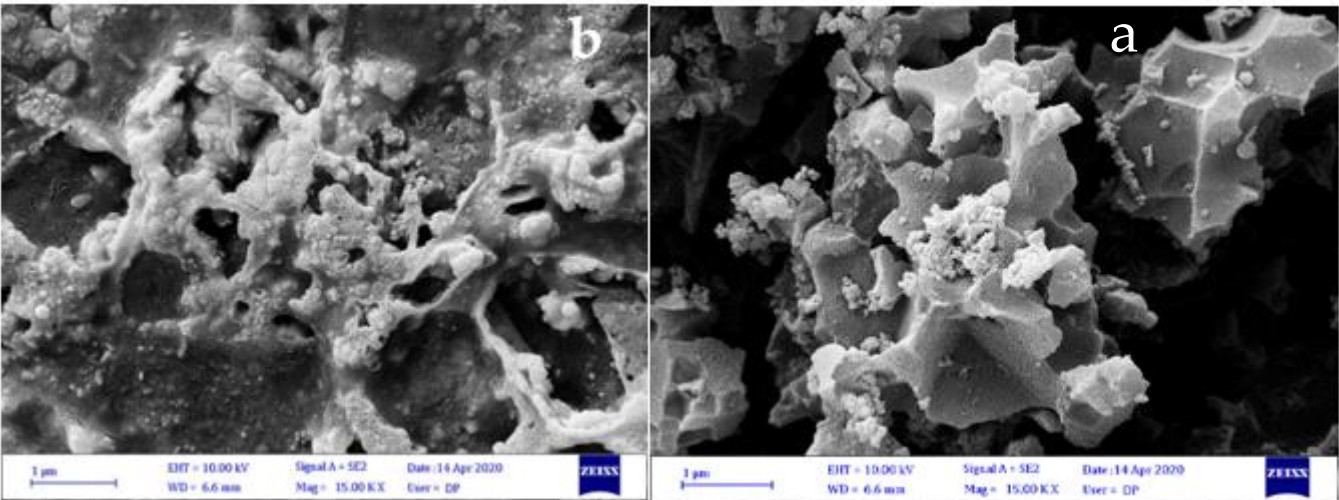

**Figure 6.** Scanning electron micrograph (SEM) of: (**a**) raw materials (walnut shells) and (**b**) the activated carbon.

**Table 2.** Obtained BET values.

| $a_s$ (m$^2$) | m (g) | $a_m$ (nm$^2$) | $v_m$ (cm$^3$ at STP, g$^{-1}$) | C | sBET (m$^2$/g) |
|---|---|---|---|---|---|
| 142.15 | 0.0329 | 0.162 | 29.16 | 212.69 | 4320.7 |

$a_s$ = surface area of the sample, m = the weight the sample in g, $a_m$ = the surface area of one molecule of N$_2$, $v_m$ = the volume of adsorbed gas at STP.

The value of C, being greater than 200, is consistent with pure activated carbon. (Values of C in the range of 2–50 indicate that the product is either a metal or polymer or an organic substance).

The specific surface area *sBET* was calculated using the following equation:

$$sBET = \frac{v_m N a \; a_m}{m \; 22400} \tag{6}$$

where *sBET* is the total surface area s per gram, *m* is the molecular weight of nitrogen, $N_a$ is Avogadro's number ($6.022 \times 10^{23}$/mole), $a_m$ is the surface area of one molecule of nitrogen, and 22,400 is the ideal gas volume.

By this measure, the specific surface area of the AC was 4320.7 m$^2$/g. A surface area of this size suggests that activated carbon would be an excellent candidate for adsorptive applications, including the adsorption of CO$_2$.

The surface area was also investigated via an iodine number, and then calculated via the ASTMD D4607−14 26 relationship [26].

$$I.N = [A − (DF \times C \times B)]/m \tag{7}$$

where A = 1269.3 according to the molarity equation M = wt × 1000/M. Wt × V, C is the volume of the burette after discoloration, B is iodine concentration in (N) multiplied by the iodine atomic weight, and DF is 2.2, which represents the coefficient factor of 100 + 10 = 110, 50 mL of which was taken out for the carburation process. At the same time, m is the weight of AC.

As can be seen in Figures 7 and 8, when the pressure against the amount of adsorbent N$_2$ gas is plotted, the appearing isotherm confirms the adsorption of nitrogen by the activated carbon. At a low pressure, the linear curve gradually begins to increase due to the adoption of N$_2$ via the activated carbon. This part of the curve represents monolayer adsorption and microfilming. The adsorption process continues until it reaches a maximum

where it stabilizes due to saturation of the activated carbon sample with $N_2$ gas. This proves that the sample of activated carbon has a high surface area. When the isotherm is reversed, in accordance with nitrogen being withdrawn, the new linear curve indicates that the mesopores behave according to the Kelvin equation, until stabilization is achieved when the mesopores are completely filled [27].

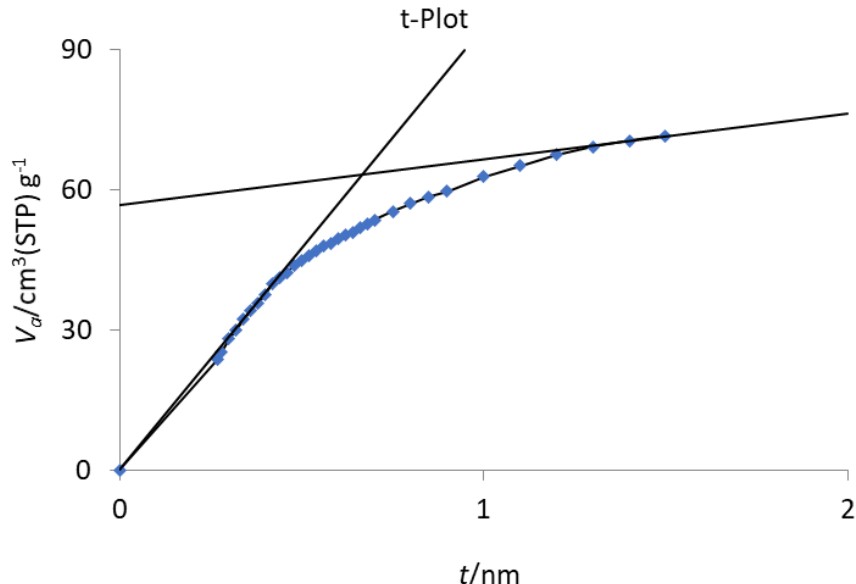

**Figure 7.** BET adsorption isotherm of $N_2$ gas on the activated carbon (ADS = adsorption, DES = desorption, STP is standard temperature and pressure 1 atm, 0 °C).

**Figure 8.** Isotherm curve, amount of adsorbent in (na mmol/gm) versus layer thickness (nm) of activated carbon (AC).

### 3.4. X-ray Photoelectron Spectroscopy (XPS)

The XPS photoelectron spectroscopy survey spectrum (Figure 9) displayed main peaks at 284.1, 284.9, 286.5, 288.0, 289.0, and 291.2 eV. The binding of C=C and C-C

corresponding to sp$^3$ and sp$^2$ carbon hybrid) have strong beaks sites at 284.1 and 284.9, respectively, while other peaks for C-H, C-O, C=O, O-C=O and $\pi$–$\pi$ interactions showed smaller peaks, meaning that around 80% of activated charcoal is caused by carbon–carbon contents [28–31].

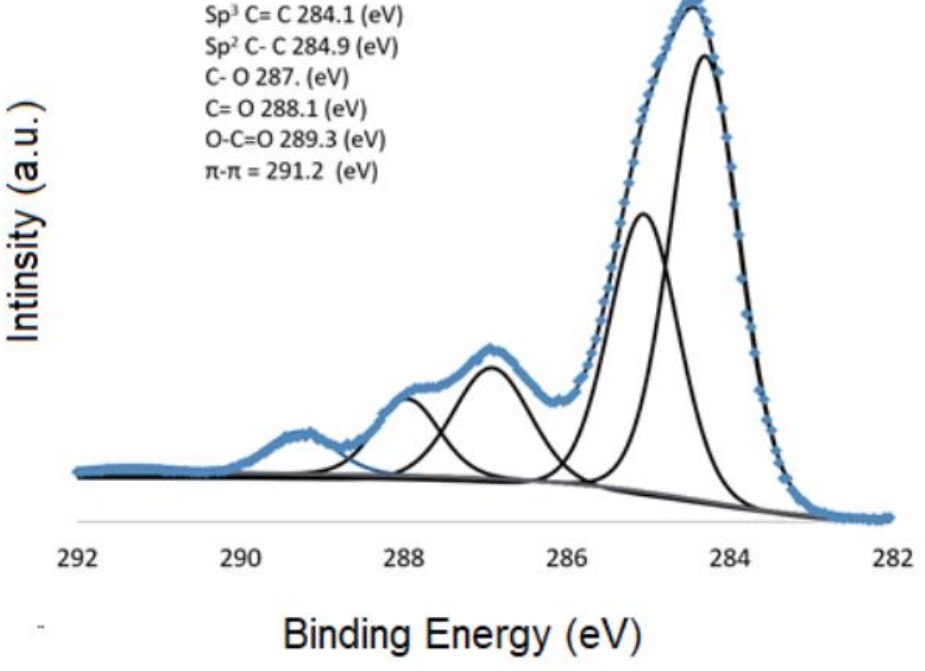

**Figure 9.** XPS spectrum of the activated carbon.

The results of the $CO_2$ adsorption investigated according to the description in Section 2.11. As can be shown in Table 3 $CO_2$ adsorption ratios of 301.1 mmol $CO_2$ per gram AC were highest adsorption value that reported, throughout first 60 min of detention time of $CO_2$ gas in AC gaps and caves. Meanwhile, no increase in $CO_2$ gas adsorption occurred when the other parameters, such as primary AC weight (0.100 g) and calcium carbonate (4 g), were fixed, which typically produced around 1.596 g of $CO_2$ gas.

**Table 3.** Results of studies on the adsorption of $CO_2$ by AC.

| Weight of AC (g) | Weight of Produced $CO_2$ (g) | Time (min) | Weight of AC and $CO_2$ (g) | Weight of Adsorbed $CO_2$ (g) | $CO_2$ Adsorption Ratio (mmol/g) |
|---|---|---|---|---|---|
| 0.100 | around 1.596 | 15 | 0.889 | 0.789 | 179.3 |
| 0.100 | around 1.596 | 30 | 1.078 | 0.978 | 222.2 |
| 0.100 | around 1.596 | 60 | 1.425 | 1.325 | 301.1 |
| 0.100 | around 1.596 | 90 | 1.418 | 1.273 | 289.3 |

Figure 10 shows the thermal stability of activated carbon that was prepared. It can be confirmed that AC can be used in simple and medium thermal conditions, as well as in other applications that require a high temperature such as catalysts in industrial chemical reactions.

Finally, Table 4 shows the most common physical properties of the prepared AC like percentage of moisture, density and ash, methylene blue absorption and iodine number of the prepared activated carbon.

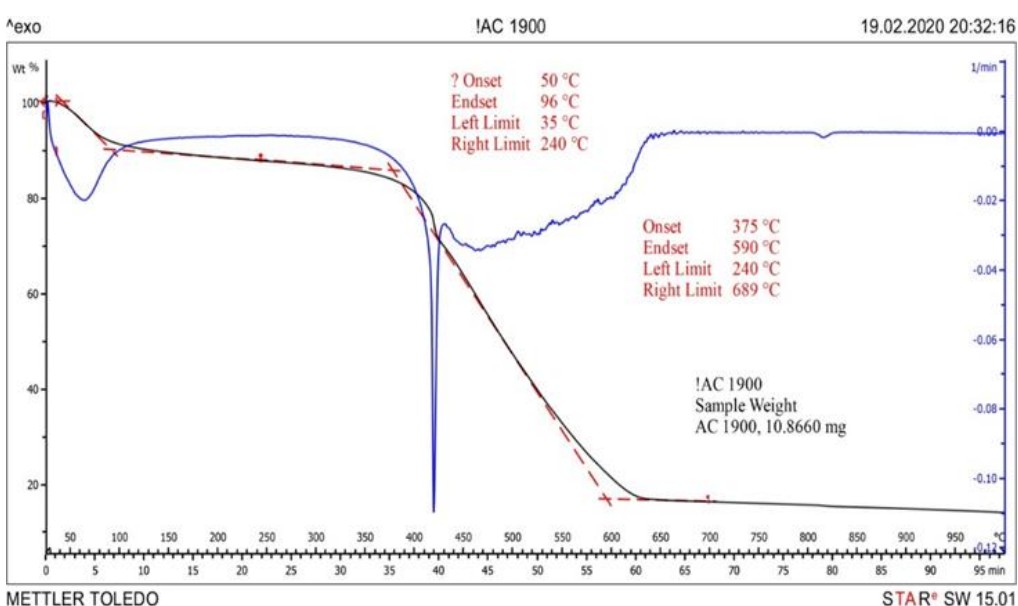

**Figure 10.** Thermogravimetric stability curve (TGA) and the differential curve (DTA) of activated carbon.

**Table 4.** Some physical properties of the prepared AC.

| Moisture% | Ash% | Density (g/cm$^3$) | Methylene Blue (mg/g) | Iodine Number (m$^2$/g) |
|---|---|---|---|---|
| 12 | 6 | 0.16 | 83 | 4126 |

## 4. Conclusions

AC with a surface area of 4320.7 m$^2$/g was prepared via a chemical and hydrothermal process. SEM images confirmed the porosity and roughness of its surface. An FTIR study of prepared AC reported decreases in the hydroxyl group (OH$^-$) and the removal of aliphatic bonds of both (C-H) and CH$_2$ and CH$_3$ groups due to thermal and chemical AC modifications. When AC was exposed to CO$_2$ for 60 min in a closed chamber at atmospheric pressure, the adsorption of a 301.1 mmol CO$_2$ (13.25 g CO$_2$) per gram of activated carbon was observed. This adsorption was a result of AC porosity and its surface roughness. While not surpassing the highest recorded adsorption ratio of 920.6 mmol/g, this is still notably high. In addition, the thermal stability of AC was stable at high temperatures, proving that AC could be useful for many industrial applications such as catalysts.

**Author Contributions:** Conceptualization, M.A. and A.S.A.; methodology, A.S.A.; software, A.A.S.; validation, M.A., A.S.A. and A.A.S.; formal analysis, M.A.; investigation, A.S.A.; resources, A.A.S.; data curation, M.A.; writing—original draft preparation, A.S.A.; writing—review and editing, G.F.S.; visualization, G.F.S.; supervision, G.F.S. All authors have read and agreed to the published version of the manuscript.

**Funding:** This research received no external funding.

**Data Availability Statement:** Data are available on request from the corresponding author.

**Conflicts of Interest:** The authors declare no conflict of interest.

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
