# Peer review of "Carbon Dioxide Adsorption by a High-Surface-Area Activated Charcoal"

_jcs, doi:10.3390/jcs7050179_

Round 1

Reviewer 1 Report (Previous Reviewer 1)

Expand and modify the abstract

Paragraph 35 AC is a black material that is solid and tasteless with highly porosity.... It goes at the beginning of the introduction, then references are given, you wrote in general, then cited examples, then returned to general, theory

Paragraph 36, which different materials?

Paragraph 41, 41 you mentioned the layers in AC in the experiment you kept for 1 hour, you probably have a turbostratic structure, so you can write something about it.

Paragraph 46, What did you mean...area particularly the edges present?

Paragraph 67, Why do you think that ash from activated carbon cannot be used, when there are many papers that talk about it

In the experimental part, paragraph 2.1, you mentioned walnut waste? do you mean the nut shell? And why didn't you state how much you used, how you performed the pretreatment

Delete paragraph 2.3 or put it in the introduction

The experiment is not an introduction, fix it and write it nicely and describe the experiment, state in detail how it was done

Did you calculate the mass loss or did you just give the formulas

Regarding the FTIR analysis, compare the raw and activated sample

Paragraph 3.2 give a more detailed analysis, why is there no SEM raw sample and then a comparison?

Why didn't you show the distribution of pores I alpha S, you didn't indicate in the table whether the sample was micro or mesoporous, you didn't indicate what type of isotherm it is

You could do an elemental analysis for the raw and activated sample and after CO2 sorption

Improve your English

Author Response

Reviewer one

Expand the abstract

It was expanding and modified

Paragraph 35 AC is a black material that is solid and tasteless with highly porosity.... It goes at the beginning of the introduction, then references are given, you wrote in general, then cited examples, then returned to general, theory

Thanes a text was added in the introduction

Paragraph 36, which different materials?

Thanks a text has added to introduction and highlighted

Paragraph 41, 41 you mentioned the layers in AC in the experiment you kept for 1 hour, you probably have a turbostratic structure, so you can write something about it.

It was clarified in the introduction

Paragraph 46, What did you mean...area particularly the edges present?

It was corrected and highlighted

Paragraph 67, Why do you think that ash from activated carbon cannot be used, when there are many papers that talk about it

AC porosity is inversely proportional with ash. while ash could some other inorganic material. We do not mean it not useful but in our perpetration less ash resulting in more active carbon.  

In the experimental part, paragraph 2.1, you mentioned walnut waste? do you mean the nut shell? And why didn't you state how much you used, how you performed the pretreatment

Thanks it was corrected

Delete paragraph 2.3 or put it in the introduction

Thanks for your comment we think the title was wrong it our experimental part .

The experiment is not an introduction, fix it and write it nicely and describe the experiment, state in detail how it was done

Thanks we fixed it

Did you calculate the mass loss or did you just give the formulas?

Thanks we fixed it

Regarding the FTIR analysis, compare the raw and activated sample

Thanks was corrected.

Paragraph 3.2 give a more detailed analysis, why is there no SEM raw sample and then a comparison?

Thanks a text was added

Why didn't you show the distribution of pores I alpha S, you didn't indicate in the table whether the sample was micro or mesoporous, you didn't indicate what type of isotherm it is

It was clarified and fixed

You could do an elemental analysis for the raw and activated sample and after CO2 sorption

It was added

Improve your English

The edit from native English speaker

Reviewer 2 Report (Previous Reviewer 3)

I write you in regards to the manuscript entitled “Carbon Dioxide Adsorption by a High Surface Area Activated 2 Charcoal, submitted to the Journal of Composite Science. Even though, the authors provided some revision, still I don’t think that the manuscript could be acceptable in present form. The detailed comments are below:

- The main problem is the serious error in calculation of the surface area. The BET adsorption isotherm of the N2 gas seem to indicate that the surface area of the AC is just more than 100 m2/g. The value of 142.15 derived from The BET adsorption isotherm of the N2 gas is the BET surface of the AC as it already calculated for the 1g of the AC whatever weight of the AC you put in the machine. So the BET surface are of the AC should be 142.15 m2/g.

- The discussion part of the work should be extended. Otherwise, I don’t this work is suitable for the publication.

- I request the author extend abstract and Conclusion part to provide more detail on the work.

Author Response

Comments and Suggestions for Authors

I write you in regards to the manuscript entitled “Carbon Dioxide Adsorption by a High Surface Area Activated 2 Charcoal, submitted to the Journal of Composite Science. Even though, the authors provided some revision, still I don’t think that the manuscript could be acceptable in present form. The detailed comments are below:

- The main problem is the serious error in calculation of the surface area. The BET adsorption isotherm of the N2 gas seem to indicate that the surface area of the AC is just more than 100 m2/g. The value of 142.15 derived from The BET adsorption isotherm of the N2 gas is the BET surface of the AC as it already calculated for the 1g of the AC whatever weight of the AC you put in the machine. So the BET surface are of the AC should be 142.15 m2/g.

We still not understand your point for this issue. we follow the BET lay to calculate surface area as was described in many paper. In equation 3 why we did not calculate the mass that put in machine. We can put more than 0.015 g to be higher surface area !!!!! . which not gave any sensitivity surface measuring!!! If you have any reference please tell us !!

- The discussion part of the work should be extended. Otherwise, I don’t this work is suitable for the publication.

Thanks It was improved

- I request the author extend abstract and Conclusion part to provide more detail on the work.

Thanks It was improved

Round 2

Reviewer 1 Report (Previous Reviewer 1)

Dear Authors,

you have corrected and modified the work according to my remarks and suggestions, therefore I agree to accept your work in the journal.

Reviewer 2 Report (Previous Reviewer 3)

Please cite me some me some reference which calculated the surface area as you did? 

This manuscript is a resubmission of an earlier submission. The following is a list of the peer review reports and author responses from that submission.

Round 1

Reviewer 1 Report

Change the title, what does the word strong in the title mean?

In the introductory part, near the end of the paragraph, you mentioned the crystal structure? since when does AC have a crystal structure?

If the crystal structure is as you say why did you not show XRD in your sample results.

Row 53, you stated that if the AC powder has a large surface area and adsorption, where did you get that from? if so please provide references.

It seems to me that you pasted equation 1, paragraph 129, write it

In paragraph 2.1, indicate which methods you used, you mentioned XPS, and then below in the results you have FTIR, SEM and BET

Paragraph 2.2 you mentioned an inert atmosphere, which one? hydrocarbons and volatiles are also removed at low temperatures.

Row 98, AC is not acid or base leached but chemically mechanically or mechanochemically treated to be chemically activated

Paragraph 3.1 line 160, you mentioned picture 4, I did not see it in the paper

In Figure 3, list the functional groups and mark the samples instead of a and b.

Paragraph 3.2 describe in more detail what you see in the picture and give a comparative analysis with the raw starting material

In paragraph 3.3, you did not provide a graph of the distribution of pores and alpha s

How and on what basis did you get table 3?

Must improve English in work

Author Response

Reviewer one

Comments and Suggestions for Authors

  1. Change the title, what does the word strong in the title mean

Response

The title has been changed

  1. In the introductory part, near the end of the paragraph, you mentioned the crystal structure? since when does AC have a crystal structure?

Response

The word " crystal " was set wrongly due to translation language differences. It was corrected (highlighted)

  1. If the crystal structure is as you say why did you not show XRD in your sample results.

Response

It was corrected in point 2

  1. Row 53, you stated that if the AC powder has a large surface area and adsorption, where did you get that from? if so please provide references.

Response

At the end of the paragraph in line 60 we wrote references (15-17)

  1. It seems to me that you pasted equation 1, paragraph 129, write it

Response 

Thanks we fixed it.

  1. In paragraph 2.1, indicate which methods you used, you mentioned XPS, and then below in the results you have FTIR, SEM and BET

Response

Thank you now was fixed

  1. Paragraph 2.2 you mentioned an inert atmosphere, which one? hydrocarbons and volatiles are also removed at low temperatures

Response

The word inert was wrongly written due to use language translation, we meant isolated environment.

  1. Row 98, AC is not acid or base leached but chemically mechanically or mechanochemically treated to be chemically activated

Response

The was wrong word replaced and highlighted. Thanks

  1. Paragraph 3.1 line 160, you mentioned picture 4, I did not see it in the paper, In Figure 3, list the functional groups and mark the samples instead of a and b.

Response                

 Thanks, it was corrected the Figure number was corrected and highlighted

  1. Paragraph 3.2 describe in more detail what you see in the picture and give a comparative analysis with the raw starting material

Response

We described thoroughly with enough details on SEM image but unfortunately, we did not do SEM for raw material to be comparable.

  • In paragraph 3.3, you did not provide a graph of the distribution of pores and alpha s

Unfortunately we did not examine EDS distribution 

Response

 Thank you was corrected

  • How and on what basis did you get table 3?

Response

The data were collected from testing system for CO2 adsorbed. then the outcome organized in this table

Reviewer 2 Report

REVIWER 2

1. What kind of surface- geometrical or specific surface?

Response

According to sem image the surface area is not specific and does not geometry .

Re: But you claim that "In granular form, activated carbon may have a particle size of >1 mm, a small diameter, a large inner surface and a relatively small outer surface."

 By which SEM method can the external and internal surfaces be determined?

4. This subsection is completely redundant. It contains a description of well-known and repeatedly described methods for obtaining and activating solid pyrolysis residues.

A text was added

 Re: Sorry, but no text is added in the revised manuscript.

 5. The description of the experimental design is unclear. You mix raw walnut shells with KOH, and place this mixture in a crucible, which you heat in a muffle furnace at 500 C, for 1h. Is this crucible open or covered? What is the atmosphere in the furnace, inert or air? You have described the general principles of obtaining an AC (Subsection 2.2) but do not follow them. Why?

 A text was added

 Re: There is no difference between the first and the revised form of the manuscript. No new text is visible that relates to my previous remarks.

Author Response

Reviewer two

  1. 1. What kind of surface- geometrical or specific surface?

Response

According to sem image the surface area is not specific and does not geometry .

Re: But you claim that "In granular form, activated carbon may have a particle size of >1 mm, a small diameter, a large inner surface and a relatively small outer surface."

 By which SEM method can the external and internal surfaces be determined?

Response

The discussion mentioned above just provided general statements about ACs.  The SEM was only to show pores and caves. While a small diameter, a large inner surface can be examined by BET characterization

  1. 4. This subsection is completely redundant. It contains a description of well-known and repeatedly described methods for obtaining and activating solid pyrolysis residues.

A text was added

 Re: Sorry, but no text is added in the revised manuscript.

Response

This has been revised

  1. The description of the experimental design is unclear. You mix raw walnut shells with KOH, and place this mixture in a crucible, which you heat in a muffle furnace at 500 C, for 1h. Is this crucible open or covered? What is the atmosphere in the furnace, inert or air? You have described the general principles of obtaining an AC (Subsection 2.2) but do not follow them. Why?

 A text was added

 Re: There is no difference between the first and the revised form of the manuscript. No new text is visible that relates to my previous remarks.

 Response

The crucible was covered. It was isolated environment. that was wrong word due to translation 

Reviewer 3 Report

I write you in regards to the manuscript entitled “Carbon Dioxide Adsorption by a High Surface Area Activated 2 Charcoal, submitted to the Journal of Composite Science. After going through the manuscript, I think that the manuscript has been quite poor in preparation with lack of scientific sound and logic. Even though, I believe that the content of the manuscript could attract some attention of the scientist in the field, the manuscript must be intensively revised before reconsidering for the publication. The detailed comments are below:

- Subsection 2.3 “General principles for the preparation of activated carbon” should move to the introduction part.

- The novelty of the work is too vague. It should be clearly stated in the introduction part.

- Discussion of FTIR should be supported with the reference to be more reliable

- The whole discussion part was too brief and general. The authors are advised to rewrite the whole discussion part of manuscript.

- The author should carefully check the surface area; the surface area of 4320.7 m2/g is too high for activated carbon. The SEM images did not show extremely high porous structure of prepared activated carbon. More importantly, the BET results (table 2) show that the specific surface area of activated carbon from walnut was 142.15 m2/g.

- Please specify the employment of Na2CO3, Na2S2O3 and KI

- It could be calculated that 1g of CaCO3 produced a maximal of 0.44g CO2. Why did the authors claim the formation of 1.596g of CO2 produced from 1g of CaCO3. Please specify.

- For the CO2 adsorption testing, the CO2 cylinder with controlled CO2 flow should be used for more reliability.

- Abstract and Conclusion part should be extended to show more results from the study.

- Please remove the citation form the conclusion part.

- More experiment on the CO2 adsorption must be included such as kinetics and isotherm of the adsorption process.

- English is quite poor. It should be extensively correct and revise.

Round 2

Reviewer 1 Report

I see that you have accepted my suggestions, my opinion is that the paper should be accepted in the journal.

Reviewer 2 Report

No comments